# The occurrence of ansamers in the synthesis of cyclic peptides

Guiyang Yao [1,2,6], Simone Kosol [1,6], Marius T. Wenz [3,6], Elisabeth Irran[1], Bettina G. Keller[3], Oliver Trapp [4,5] & Roderich D. Süssmuth [1] ✉

α-Amanitin is a bicyclic octapeptide composed of a macrolactam with a tryptathionine cross-link forming a handle. Previously, the occurrence of isomers of amanitin, termed atropisomers has been postulated. Although the total synthesis of α-amanitin has been accomplished this aspect still remains unsolved. We perform the synthesis of amanitin analogs, accompanied by in-depth spectroscopic, crystallographic and molecular dynamics studies. The data unambiguously confirms the synthesis of two amatoxin-type isomers, for which we propose the term ansamers. The natural structure of the *P*-ansamer can be ansa-selectively synthesized using an optimized synthetic strategy. We believe that the here described terminology does also have implications for many other peptide structures, e.g. norbornapeptides, lasso peptides, tryptorubins and others, and helps to unambiguously describe conformational isomerism of cyclic peptides.

The chemical synthesis of constrained peptide macrocycles of natural origin or of designed artificial peptides sometimes leads to the occurrence of isomers, which have been designated with various terms. There exist various literature reports: Wareham et al. describe the homeomorphic isomerism of macrobicyclic peptidic compounds which involved a passage of the bridge chain through the macrolactam (Fig. 1a)[1]. Bartoloni et al. investigated the diastereomeric norborna-peptides as potential drug scaffolds which showed bridge-up/down orientations according to the NMR solution structure (Fig. 1b)[2]. The Yudin group reported an unusual tunable atropisomeric peptidyl macrocycle which is made possible by controlling the conformational interconversion[3]. More recently, Baran and co-workers accomplished the reported total synthesis of the peptidic indole alkaloid tryptorubin A and defined non-canonical atropisomers, a family of shape-defined molecules that are distinguished by bridge below/bridge above arrangements (Fig. 1c)[4]. However, the vernacular nomenclature incited some controversy and Crossley and co-workers suggested a composite phenomenon using polytope formalism which is the fundamental of akamptisomerism classification[5]. Ultimately, the existence of

atropisomers has also been postulated for the peptide toxins phalloi-dins and amanitins[6–12].

Phallotoxins and amatoxins are two bicyclic peptide toxin families isolated from the death cap mushroom *Amanita phalloides*. They both belong to the ribosomally synthesized and post-translationally modified peptides (RiPPs) and display high toxicity with low lethal doses in vivo animal experiments. α-Amanitin **1**, a slow acting toxin ($LD_{50}$ = 50 – 100 μg/kg), has been reported to be a selective inhibitor of RNA polymerase II[13,14]. Its bicyclic octapeptide structure contains a 6-hydroxy-tryptathionine-(*R*)-sulfoxide cross-link (Fig. 1d). With the macrolactam ring as an imaginary plane, in a typical presentation, the tryptathionine bridge is located as a handle above the macrolactam, as it can be derived from a previously published X-ray structure (Fig. 1e)[15].

Longstanding questions are whether so-called atropisomers indeed existed and if so, under which circumstances they would occur, and what type of isomerism this would be? Previous studies of surro-gate molecules of phalloidin and amanitin reported NMR spectro-scopic data accompanied by CD spectroscopic analysis[8] and molecular dynamics (MD) simulations[7]. However, epimerization of sidechain

[1]Institut für Chemie, Technische Universität Berlin, Strasse des 17. Juni 124, 10623 Berlin, Germany. [2]Center for Innovative Drug Discovery, Greater Bay Area Institute of Precision Medicine (Guangzhou), School of Life Sciences, Fudan University, Shanghai, PR China. [3]Department of Biology, Chemistry, Pharmacy, Freie Universität Berlin, Arnimallee 22, 14195 Berlin, Germany. [4]Department of Chemistry and Pharmacy, Ludwig-Maximilians-University, Butenandtstr. 5-13, 81377 Munich, Germany. [5]Max-Planck-Institute for Astronomy, Königstuhl 17, 69117 Heidelberg, Germany. [6]These authors contributed equally: Guiyang Yao, Simone Kosol, Marius T. Wenz. ✉e-mail: suessmuth@chem.tu-berlin.de

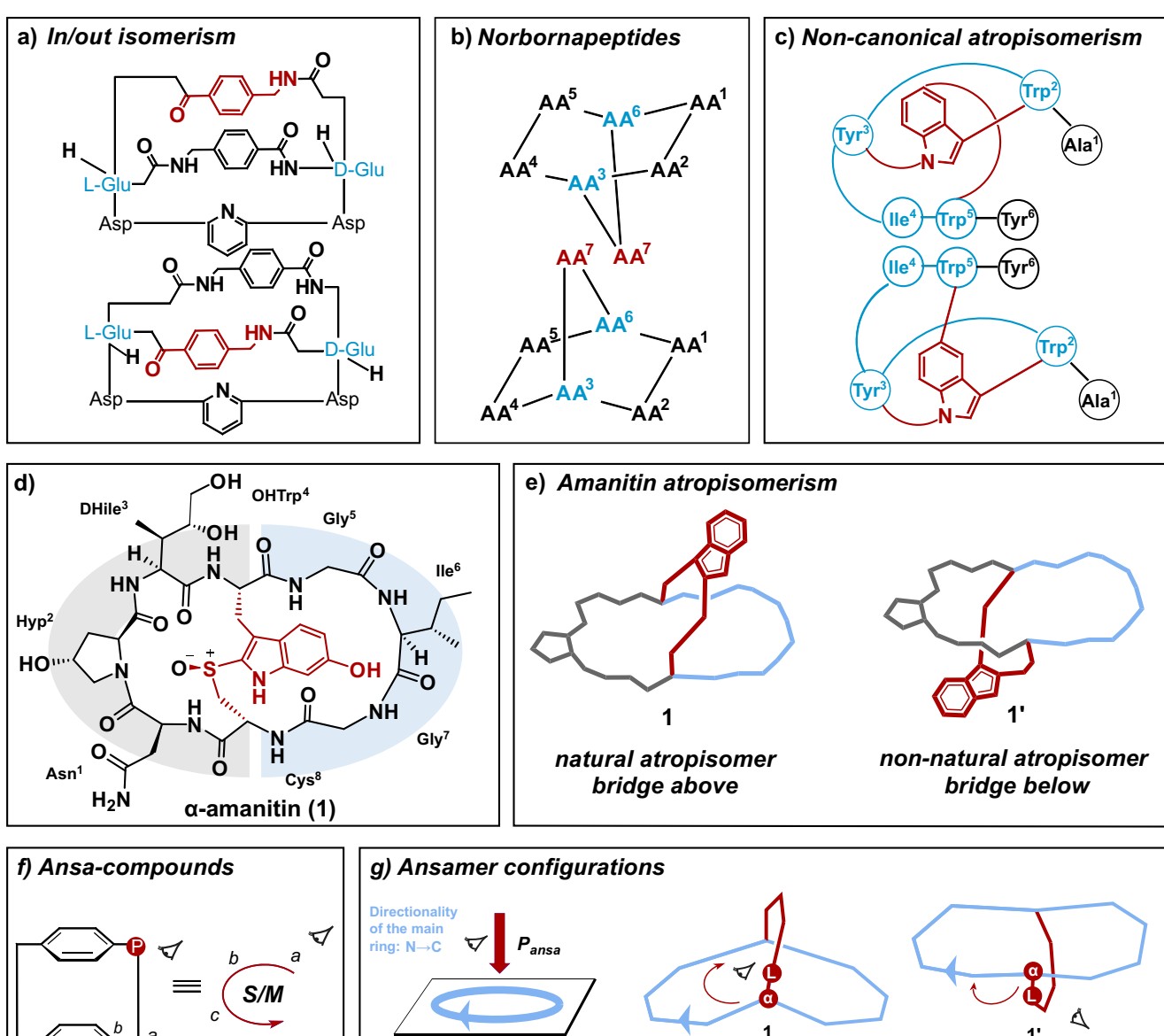

**Fig. 1 | Peptide isomerism types and suggested determination of the ansamer configuration of cyclic amanitin-type systems.** Several nomenclatures have been proposed to describe the isomerism of peptides displaying ring systems (**a**–**c**). **d** Structure of α-amanitin with the tryptathionine bridge highlighted in red. DHIle (2 S,3 R,4R-dihydroxyisoleucine), Hyp (4-*trans*-L-hydroxyproline), OHTrp (6-OH-tryptophan). **e** Proposed conformational isomers of α-amanitin. The A and B-rings are colored in gray and blue, respectively. **f** Determination of the

configuration for ansa-compounds[24]. **g** Determination of ansamer-configurations: (1) the main cycle and the directionality is identified (light blue): from N- to C-terminus. (2) Bridge-up/bridge-down cases in view of the priority order in the main cycle. Identification of the leading atom/group L of the bridge next to the bridgehead atom α is assigned. (3) The descriptor $P_{ansa}$ or $M_{ansa}$ is assigned according to the directionality (clockwise/counter-clockwise) from the position of the leading atom/group L.

stereocenters as an alternative explanation could not be unambiguously ruled out[6].

Here, in a systematic approach combining various analytical methods (Marfey analytics, X-ray crystallography, NMR, and CD spectroscopy) with MD simulations, we determine the structure and dynamics of this sought isomer. Furthermore, we investigate different macrolactamization sites and show that an optimized strategy can ensure atroposelective synthesis. Finally, we propose the term ansamer to describe and unambiguously assign the configuration of stereoisomers of bridged cyclic systems, which can exist as configurational stereoisomers, depending on the position of the bridge, above or below the main ring. We suggest applying this

terminology also to other conformationally restricted cyclic peptides, such as norbonapeptides or lasso peptides.

## Results and discussions

### Site-dependent macrolactamization and cycloisomer formation

Apart from semi-synthetic attempts, four total syntheses of α-amanitin have been reported to date. These contain three basic synthetic approaches to install the characteristic tryptathionine: An initial route employed the Savige-Fontana reaction via an Hpi (3a-hydroxy-pyrrolo[2,3-b] indole)[16] intermediate by the team from Heidelberg Pharma GmbH. This reaction was also used in a more sophisticated fashion by Perrin and co-workers to accomplish the first total synthesis of

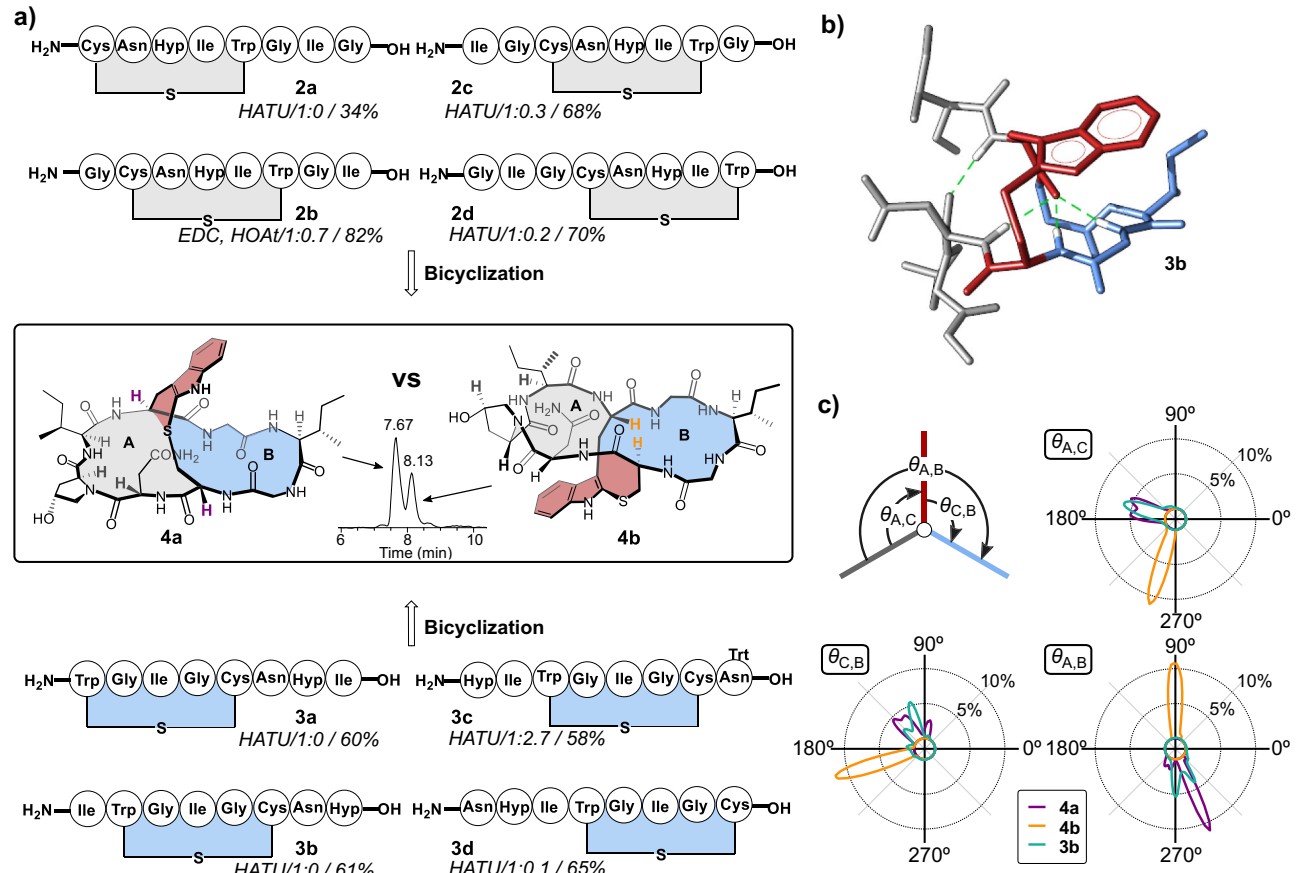

**Fig. 2 | Peptide precursor choice impacts isomer yields. a** Bicyclization conditions (HATU or EDC/HOAt) and different ring closure sites with preformed A-ring **2a-d** (gray) or B-ring **3a-d** (blue) lead to different isomer ratios (**4a**:**4b**) and yields (**4a**+**4b**). **b** MD-simulated structure of precursor **3b** with highest probability. Hydrogen bonds are shown as green dashed lines. The structure is colored according to the scheme in Fig. 1e. **c** Illustration of the model system to assess the relative orientation of the A-ring, B-ring, and tryptathionine bridge in the amanitin scaffold. The three defined planes (A-ring in gray, B-ring in blue and tryptathionine bridge in red, as in **a**, **b** are shown as lines, their relative orientation is defined by the angles θ as indicated (top left). For each angle ($\theta_{A,C}$, $\theta_{C,B}$, and $\theta_{A,B}$), the average distribution over 20 μs simulation data is presented in a circular plot. The distributions of angles observed in simulations of **4a** (violet), **4b** (orange), and **3b** (cyan) are shown as lines.

amanitin[17]. One approach from our lab was using a preformed tryptathionine from the reaction of indoles with sulfenyl chlorides, in a convergent [5 + 1 + 2] synthesis strategy[18]. Recently, we developed a robust and versatile iodine-mediated tryptathionine formation protocol that enabled us to synthesize various amanitin analogs for detailed SAR studies[19]. When we used the protocol to sample different macrolactamization sites (Fig. 2a), we noticed in some cases the formation of a by-product with an identical molecular mass as the desired amanitin analog. This could be interpreted as the formation of a diastereomer. A preliminary Marfey analysis[20] could however not prove epimerization, which led us to assume another isomer effect.

To establish a simplified model system to further investigate this observation, we decided to use readily available amino acids. Therefore, in the pursuit of the synthetic amatoxin we replaced DHIle[3] with Ile[3] to obtain Ile[3]-S-deoxo-amaninamide (Fig. 2). Thus, eight overlapping monocyclic peptides (**2a–2d** and **3a–3d**) were synthesized following our previous work[19] and the subsequent final macrocyclization was performed by using either HATU or EDC/HOAt as coupling agent, rendering eight corresponding bicyclic peptides. For the bicyclization of **2b**, the coupling reagent HATU was initially tested but significant amounts of the guanidination product were detected. To suppress the formation of the guanidination product, EDC/HOAt was selected as coupling reagent. Since the ratio of **4a** and **4b** is not changed much under different coupling conditions (see Supplementary information section 3.1.8.2 and Supplementary Fig. 19), we

concluded that different bicyclization reagents and conditions do not significantly change the ansa-selectivity of the reaction. The crude peptides were carefully analyzed by HPLC-MS (see Supplementary Fig. 1). The synthesis route via monocyclic peptides containing ring A (**2a–2d**), repeatedly resulted in two peaks (**4a** and **4b**, $R_t$ = 7.67 min and 8.13 min, respectively; Supplementary Fig. 1). Both peaks had the identical molecular mass of bicyclic Ile[3]-S-deoxo-amaninamide ([M + H]⁺ = 855.3851 Da), albeit occurring at different ratios (1:0 to 1:0.7, see Fig. 2a). In contrast, for precursor peptides with monocycle B as intermediate (**3a-3d**) only one peak was observed, with the exception of **3c** which favored the formation of **4b** (ratio 1:2.7, Fig. 2a). When screening different macrolactamization sites, the ratio of isomer yields did not follow a clear trend. Precursors with preformed A-ring (**2a–2d**) tended to result more often in isomeric product mixtures, while precursors with preformed B-rings (**3a-3d**) mostly yielded the natural isomer. The NMR spectra of the two isomers formed by different macrolactamization strategies correspond to each other which also excluded an epimerization during bicyclization (see Supplementary Figs. 14 and 15).

In our established iodine-mediated tryptathionine formation protocol, we employed precursor **3b** which exclusively yields **4a** (Fig. 2a). To investigate if **3b** exhibits conformational pre-organization favoring **4a**, we performed classical MD simulations of the precursor as well as of **4a** and **4b**. Indeed, in the simulations, **3b** forms a stable hydrogen bond network (Fig. 2b, Supplementary Fig. 2b), which allows

the molecule to adopt a conformation in which the C- and N-termini are spatially close and the tryptathionine bridge is located above the B-ring (Fig. 2b, Supplementary Fig. 3d). To compare the structural organization of **3b** with **4a** and **4b**, we defined three planes and the angles ($\theta$) between them: one plane for ring A and B, respectively, and a third plane for the tryptathionine (Fig. 2a, c). We measured the angles in the MD simulations of **3b**, **4a** and **4b** to compare the orientation of the bridge relative to the macrocycle in the three molecules. Satisfyingly, in **3b**, the tryptathionine and the ring planes are positioned as in **4a** with angles between 90° and 180° (Fig. 2c, Supplementary Fig. 3d). This likely explains the selective reaction of monocyclic **3b** to bicyclic **4a**.

### Analytical characterization of bicyclic amaninamide isomers

Since UV absorption of tryptathionines is highly distinctive, it has been previously employed to characterize tryptathionines[21]. Interestingly, the UV maximum absorption of isobaric **4a** and **4b** is slightly different with $\lambda = 289$ nm and $\lambda = 293$ nm, respectively (Fig. 3a). To clearly exclude epimerization during bicyclization, enantiomer analysis of the amino

acids by Marfey's method[20] showed identical configuration for every amino acid in both isomers **4a** and **4b** (see Supplementary Fig. 4).

Further analysis revealed that the CD and NMR spectroscopic data of peptide **4a** are fully consistent with Ile[3]-S-deoxo-amaninamide as previously characterized[15]. In contrast, the CD and NMR spectra of **4b** are noticeably different (Fig. 3b, Supplementary Fig. 13) and suggest a different 3D structure. As reported previously for amanitin analogs[8], this could be consistent with the formation of conformational isomers. For compound **4a**, a positive Cotton effect was observed between $\lambda = 210$ nm and 230 nm (in accordance with reported CD data of the natural conformer). In contrast, the potential non-natural conformer **4b** shows a negative Cotton effect at these wavelengths (Fig. 3b).

EXSY NMR analysis indicates that **4a** and **4b** are not readily exchangeable conformers. Interconversion does not occur at elevated temperatures in DMSO (150 °C, 10 h; Supplementary Fig. 17), or water (CD spectroscopy: cycle 20 °C→90 °C→20 °C) which is consistent with VT-NMR measurements (see Supplementary Figs. 5 and 6). Interconversion of the two isomers would require the indole sidechain to pass through the macrolactam ring which appears sterically

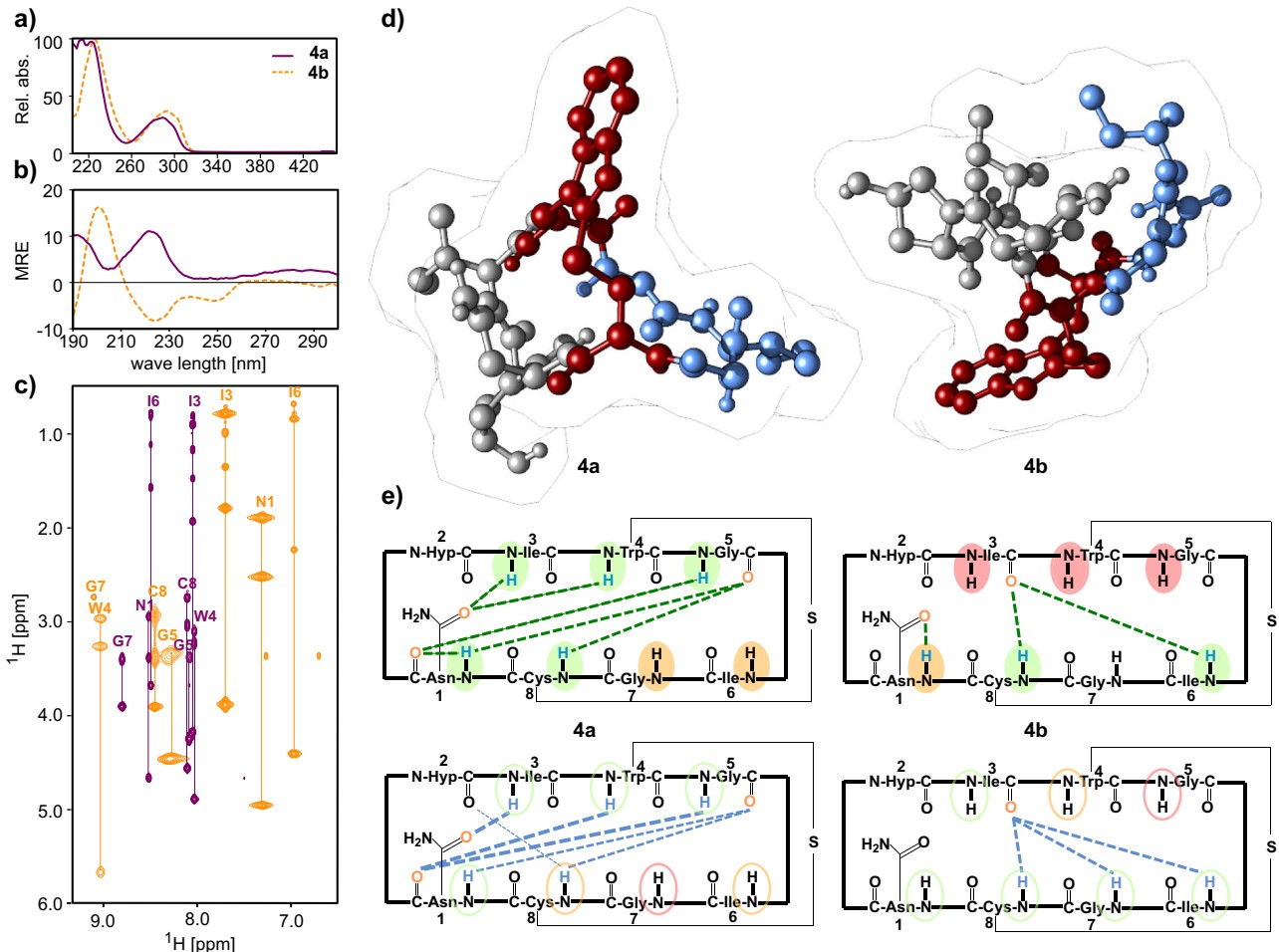

**Fig. 3 | Physicochemical differences of amaninamide isomers.** UV-absorbance spectra **a** show different maxima at $\lambda = 289$ nm and 293 nm for **4a** (purple) and **4b** (orange). **b** CD spectra of **4a** (purple) and **4b** (orange) show opposite Cotton effects at 225 nm. MRE = mean residue ellipticity in $10^3$x [deg*cm²*dmol⁻¹]. **c** Differences in chemical shifts of **4a** (purple) and **4b** (orange) in the amide region of the ¹H-¹H-TOCSY NMR spectra. **d** Crystal structures of **4a**[15] and **4b**. The A-ring and B-ring are colored gray and blue, respectively. The tryptathionine bridge is shown in red. **e** H-bonds found in the crystal structures of **4a** (left) and **4b** (right) are indicated by dashed green lines (top schematics). Amides with small chemical shift changes in VT-NMR measurements ($\Delta\delta$HN/$\Delta$T > −3.0 ppbK-1) indicative of H bonding are

highlighted in green, solvent-exposed amides with large shift changes ($\Delta\delta$HN/$\Delta$T < −4.6 ppbK-1) are highlighted in red and amides with weak shielding/H-bonding are shaded in orange (top schematics). H-bonds observed in MD simulations are shown as blue dashed lines with the line width indicating the average population over 20 μs simulation time (bottom schematics; only H-bonds that occurred in >10% of the populations are shown, see Supplementary Fig. 12). In the MD simulations, amide groups highlighted with green ellipses showed small calculated solvent accessible surface areas (SASA < 0.02 nm²), amides with large SASA (> 0.04 nm²) are circled in red and amides with intermediate accessibility are circled in orange (exact values are listed in Supplementary Table 4).

impossible without the breakage and reformation of covalent bonds (Fig. 3). This is supported by our MD simulations where we also do not observe a transition between **4a** and **4b** (Fig. 2c), even at elevated temperatures.

To further characterize the relationship between isomers **4a** and **4b** both compounds were desulphurized with Raney Nickel at 80 °C to give the corresponding macrolactams (see Supporting Information). LC-MS analysis of the paralleled reactions indicated that the monocycles **5** ($R_t$ = 5.88 min; [M + H]$^+$ = 825.4253 Da) are structurally identical (Supplementary Fig. 7: LC-MS after desulfurization). This was further confirmed by NMR spectroscopy of HPLC-purified **5** (Supplementary Fig. 8: calculated structure), suggesting that the tryptathionine bridge is the key factor of isomer formation.

Final proof for the isomeric nature of **4a**[15] (CCDC deposition number: 1128063) and **4b** was obtained from X-ray crystallography. Crystals of compound **4b** were grown in 10% EtOH aqueous solution and the structure of the peptide was obtained (CCDC: 2153904, Fig. 3d and Supplementary Fig. 9). In the crystal of **4b**, the tryptathionine bridge is clearly located below the plane of the macrolactam. The configuration of all amino acids in **4b** is identical with **4a**, corroborating our results from Marfey's analysis that no epimers were formed during macrocyclization. Remarkably, in the X-ray structure of **4b**, the trans-amide bond between Asn$^1$ and Hyp$^2$ is flipped to a cis-amide conformation with the carbonyl-group of Asn$^1$ facing towards the outside of the macrolactam instead of the inside (the trans character of the hydroxy-group of Hyp$^2$ is maintained). Overall, the backbone geometry and H-bonding pattern are drastically different from **4a**, giving the molecule a more compact appearance (Fig. 3e, Supplementary Fig. 16). This promotes the formation of a hydrophobic patch by Ile$^3$ and Ile$^6$ in **4b**, whereas in **4a** these are oriented in opposite directions. These conformational differences also substantially alter the physicochemical properties of the bicyclic peptides: hence the isomer **4b** is insoluble in water at 2 mM (Supplementary Fig. 10).

Besides the differences between crystal structures of **4a** and **4b**, which are also reflected in the angle populations shown in the Ramachandran plots (Supplementary Fig. 2a), the MD simulations also reveal differences in the dynamic behavior of the two isomers. We calculated the atomic RMSF for the simulated structures of **4a** and **4b** compared to their respective crystal structure (Fig. 3d, Supplementary Fig. 3). The RMSF values suggest that **4b** is generally more rigid than **4a** which is in line with the backbone angle distributions of **4a** being broader than of **4b** (Supplementary Figs. 2a and 3). Interestingly, quantum mechanical calculations (TDDFT level) on 100 optimized structures out of each MD data set showed that **4b** is also energetically favorable compared to **4a** (difference -30 kJ/mol).

The MD simulations and the crystal structures are in very good agreement, as the all-atom RMSD is <0.4 nm and the backbone RMSD <0.2 nm (Supplementary Fig. 21). However, we noted some differences between the hydrogen bond patterns observed in the crystal structures and the simulated structures (Fig. 3e). For **4a**, in the majority of trajectories, the hydrogen bond pattern matches the crystal structure (Supplementary Fig. 11). Interestingly, in the MD simulations, we observed a subset of structures with a different set of hydrogen bonds compared to the crystal structure. Correlation analysis of the H-bonds in this subset and in the crystal structure (Supplementary Fig. 2b) shows that they are mutually exclusive, suggesting that a different minor conformation could exist which we did not observe in our experiments. Overall, the MD simulations suggest that the structure **4a** adopted in the crystal is likely the most stable conformation of this isomer (Supplementary Fig. 11, Supplementary Table 3), which is supported by the very good agreement in the NOE distances between the NMR and MD ensembles (Supplementary Fig. 13c).

For **4b**, a bifurcated backbone hydrogen bond is present in the crystal between the carbonyl oxygen of Ile$^3$ and amide hydrogens on the opposite site of the peptide ring (Ile$^6$ and Cys$^8$). MD simulations suggest a trifurcated H-bond instead, also involving the amide of Gly$^7$ (Fig. 3e, Supplementary Fig. 12). Unfortunately, it was not possible to determine if the amide proton of Gly$^7$ is shielded by a H-bond in solution as it was not resolved in the VT-NMR experiments. However, the calculated solvent-accessible surface areas of the amide groups in the crystal and MD structures agree well with each other (Fig. 3e, Supplementary Fig. 12d, Supplementary Table 4). As in **4a**, the NMR and the MD ensembles are in very good agreement with each other (Supplementary Fig. 13c).

## Ansamers – a concept for assigning conformational isomers of cyclic peptides

Our structural and physicochemical characterization of the two isomers **4a/4b** shows that the isomers **4a/4b** are not enantiomers, but diastereomers, lastly not only because of the pronounced differences in bond geometries, but also due to the chirality of amino acids. Previously, isomers arising from differing bond geometries at the bridgehead in bicyclic peptide systems have been categorized as atropisomers[22], non-canonical atropisomers[4] or akamptisomers[5]. This has led to some controversy and at this time there is no accurate and simple term to describe such a pair of isomers[23]. Atropisomers are clearly defined as stereoisomers, which are interconverted by rotation of a single bond between connecting moieties, which are typically sterically hindered. The herein described bridged cyclic peptide structures **4a** and **4b** are different from such atropisomers, because the interconversion of these stereoisomers is not just attributed to the rotation around a hindered single bond, but a flipping of the bridged cyclic structure. For that matter, the hindered bond-angle inversion that leads to akamptisomerization appears to be a better descriptor of the observed isomerism. But in case of **4a** and **4b** it is planes instead of bonds that undergo an angle inversion (Fig. 2c) and application of heat will not convert one diastereomer to the other. In addition, the flipping of the bridged cyclic structures leads to a conformational change in the cycle. As we find that **4a** and **4b** cannot readily interconvert even under heating we would assign the isomers the same molecular formula and same bond connectivities to configurational rather than conformational isomers. We therefore propose the term ansamer to stereochemically describe the two bridged isomers **4a** and **4b**. Compared to ansa-compounds, which consist of bridged planar chiral phenylene systems[24] (Fig. 1f), in the here presented ansamers, the bridged main cycle is structurally strained (ring strain or a clamp bridge), which increases the interconversion barrier between the two isomers. In contrast to atropisomers, ansa-compounds (lat. ansa = handle), e.g. cyclophanes, are interconverted by rotation of the handle around the planar phenylene. Similar to ansa-compounds[23] the assignment of the stereochemical descriptor can be made as follows in agreement with the CIP rules[25,26]: (1) identification of the main cycle[27–29] with the preferred directionality (Fig. 1g, from N to C terminus), (2) assignment of the leading atom/group of the bridge (Fig. 1g, leading atom/ group L) next to the bridgehead atom (α), followed by (3) assignment of the priority from the position of L: clockwise/counter-clockwise sense of the main cycle (Fig. 1g). (4) The descriptors $P_{ansa}$ or $M_{ansa}$ can be assigned accordingly (Fig. 1g). These assignment rules are unambiguous and correctly describe existing enantiomers, epimers, and diastereomers. This procedure would attribute conformational isomer **4a**, to the $P_{ansa}$ and the non-natural isomer **4b** to the $M_{ansa}$ isomer. Hence, the biosynthesis of α-amanitin **1**, is $P_{ansa}$-selective in establishing the tryptathionine bridge, as only the isomer with the indole above the ring has been found in peptides from the Amanita mushroom family. A member of the flavoprotein monooxygenase (FMO) family has been suggested to catalyze C-S bond formation[30], however, little is known about this step while the cleavage of the leader peptide and cyclization by a prolyl oligopeptidase (POPB) are well characterized[31,32]. The herein proposed $P_{ansa}/M_{ansa}$ nomenclature could be also applied to previously described norbornapeptides by

Reymond et al.[2] and in an extended version, it may prove helpful to describe the conformation of in/out-isomers[1,33], the lasso peptides[34,35] and even tryptorubin[4] (see Supplementary Fig. 18).

Inspired by the terminology used for ansa-compounds, the term ansa (handle) illustrates the cause of the isomerism and at the same time reflects the planar aspect shared with benzene ansa-compounds. In this sense the nomenclature is consistent. We hope the intended terminological similarity between ansa compound for aromates and ansamer will raise attention and spur discussions in the scientific community about the underlying problem to accurately categorize this type of isomerism.

In summary, we have proven the occurrence of isomers in the synthesis of bicyclic amanitin analogs, which have been previously postulated and termed as atropisomers. The crucial step is the macrolactamization of monocyclic tryptathionine-containing peptides. The resulting isomers which appear fixed in differently bridged conformations, have been thoroughly characterized by spectroscopic and crystallographic methods as well as by MD simulations. For steric reasons, the indole sidechain of the tryptathionine cannot thread through the macrolactam ring. In a stereochemical description of these stereoisomers, we devise the term ansamers which can unambiguously describe these isomers and which can be applied also to various other cyclic peptides.

## Methods

### General procedure for the monocyclic peptide synthesis
2-CTC resin (1 g, 0.98 mmol/g) was pre-swollen for 20 min in DCM in a manual solid phase peptide synthesis vessel (10 mL). After the solvent was drained, the first amino acid Fmoc-AA[1]-OH (0.3 mmol) and DIPEA (0.26 mL, 1.5 mmol) in DCM (5 mL) were added to the resin. The mixture was agitated for 2 h and before the solvent was drained. The resin was rinsed with DMF (4 × 3 mL). Then a mixture of MeOH/DIPEA/DCM (1:1:8) was added to cap the remaining 2-chlorotrityl chloride on the resin. The mixture was agitated for 0.5 h. Then the solvent was drained and the resin was washed with DMF (4 × 3 mL). The resin loading was determined to be 0.30 mmol/g. The Fmoc-group was removed with 20% piperidine in DMF solution. Fmoc-AA[2]-OH (4 eq) was coupled to the deprotected resin according to TBTU mediated coupling. The Fmoc-group of the resulting resin was removed employing 20% piperidine in DMF. The following six amino acids were coupled to the deprotected. The tryptathionine formation was carried out on the solid support using $I_2$-mediated thioether formation. After removal of the Fmoc-group and followed cleavage from the resin, the monocyclic peptide was obtained following subsequent HPLC purification. The synthesis and characterization data of all compounds have been reported in the supplementary information.

### General procedure for the bicyclic peptide synthesis
Monocyclic octapeptide **2a**–**2d** and **3a**–**3d** (1.0 eq) was dissolved in DMF (1 mM). Then, DIPEA (2.2 eq) and HATU (2.0 eq) were added at 0 °C. The reaction mixture was allowed to warm to r.t. for 12 h and concentrated under reduced pressure. The crude product was purified using preparative HPLC to afford bicyclic octapeptide as a white powder. Since large amounts of guanidinylation product were detected during macrolactamization of **2b**, the alternative coupling condition EDC (2 eq) and HOAt (2 eq) was employed to cyclize the monocyclic peptide **2b**. All results of LC-MS runs are shown in Supplementary Fig. 1. In addition, the detritylation was performed after macrolactamization of **3c**. The yield and ratio of **4a** and **4b** is shown in Fig. 2.

### NMR assignment and structure calculation of desulfurized macrolactam
To obtain resonance assignments for NOE assignment and structure calculations **4a**, **4b**, and desulfurized macrolactam **5** were dissolved in deuterated DMSO-$d_6$ (~10 mM). TOCSY, COSY, NOESY, and

$^1$H-$^{13}$C-HSQC spectra were recorded on a Bruker Avance III 700 MHz spectrometer with a TXI 5 mm probe. Standard Bruker pulse programs were used and all spectra were acquired at 298 K. Residual solvent methyl peaks (DMSO-d7 $\delta$ = 2.502 for $^1$H and $\delta$ = 39.0 ppm for $^{13}$C) were used for chemical shift referencing. 2D homonuclear spectra were measured with acquisition times of 70 and 18 ms for the direct and indirect dimensions, respectively. TOCSY and NOESY spectra were accumulated with 16 or 32 (in case of **4b**) scans and COSY spectra with eight scans. The TOCSY and NOESY mixing times were set to 100 and 300 ms, respectively. Natural abundance $^1$H-$^{13}$C-HSQC spectra were measured with 140 scans and acquisition times of 14 and 120 ms for the direct and indirect dimensions. The spectra were processed and analyzed using TopSpin 3.5 (Bruker) and CcpNmr 2.3.1[36]. After shift assignment (Supplementary Table 1), the NOE correlations of **4a** and **4b** were manually assigned and residue interaction matrices of **4a** and **4b** were generated using CcpNmr. For structure determination, the manually assigned chemical shifts of the desulfurized macrolactam and NOESY peak lists were supplied to CYANA for automated NOE assignment and structure calculation (Supplementary Table 2). The program CYLIB[37] was used to generate a CYANA library file for 4-hydroxyproline. A set of 1000 structures was calculated and the 100 best were visually inspected with UCSF Chimera[38].

### Molecular dynamics simulations
All-atom classical MD simulations of peptides **4a**, **4b**, and **3b** in explicit dimethylformamide were carried out in GROMACS[39–41]. The force field parameters for the peptides were obtained with ACPYPE[42]. The simulations were conducted at the $NpT$ ensemble with $p = 1$ bar and $T = 300$ K or $T = 400$ K. The simulation time was 20 μs for each system at $T = 300$ K, and 0.1 μs for each system at $T = 400$ K. See SI for a detailed protocol.

### Reporting summary
Further information on research design is available in the Nature Research Reporting Summary linked to this article.

## Data availability
All processed data that support the findings of this study are available within the article and its Supplementary Information (experimental details; synthetic procedures; X-ray diffraction, NMR, UV/Vis, VT-NMR, MD simulations). All information to redo the MD simulations including topology, starting, and structure files are stored on Zenodo [DOI: 10.5281/zenodo.6974777] together with the highest-probability structures or simulated crystal structures of **4a**, **4b**, **3b**, and **3c** as well as downsampled trajectories with a visualization state for the open-source program VMD. The highest-probability structures for **4a**, **4b**, **3b**, and **3c** are provided as source data file alongside the manuscript. The X-ray crystallographic data for **4b** was deposited at the Cambridge Crystallographic Data Centre (CCDC) under deposition number CCDC 2153904 [DOI: 10.5517/ccdc.csd.cc2b99sr]. The open-source software used in this study is available under: MD simulations and analysis: GROMACS 2019.4 and GROMACS 2020.6 (https://manual.gromacs.org/documentation/), Custom code: Python 3.9.2 (https://www.python.org/downloads/), Jupyter (IPython 8.5.0, notebook version 6.4.12, https://jupyter.org); Visualization: VMD for MACOSXX86_64, version 1.9.4a57 (April 27, 2022, https://www.ks.uiuc.edu/Research/vmd/). MD data is stored together with custom code according to DFG regulations, and they are available from B.G.K. (bettina.keller@fu-berlin.de) upon request. Other data is available from the corresponding author. Source data are provided with this paper.

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

## Acknowledgements

Dr. Guiyang Yao thanks the Alexander von Humboldt Foundation for a postdoctoral fellowship and the National Natural Science Foundation of China (82204189). The work was funded by the Deutsche Forschungsgemeinschaft (DFG, German Research Foundation, RTG 2473

Bioactive Peptides to R.D.S., M.T.W., and B.G.K.). M.T.W. and B.G.K. thank the Paderborn Center for Parallel Computing PC2 and ZEDAT of FU Berlin for computing time. Molecular graphics and analyses were performed with UCSF Chimera, developed by the Resource for Biocomputing, Visualization, and Informatics at the University of California, San Francisco, with support from NIH P41-GM103311. The authors thank Manuel Gemander and Hengshan Wang for helpful discussions and critical suggestions.

## Author contributions

G.Y, S.K., and R.D.S. designed the experiments. G.Y. performed the synthesis of all shown compounds. S.K. and G.Y. designed and supervised the CD, NMR, and X-ray studies. E.I. recorded the crystallographic data and solved the structure. O.T. made contributions to the isomery interpretation and to the ansamer nomenclature. S.K., M.T.W., and B.G.K. performed the theoretical calculations of the precursor, ansamers, and macrolactam. All authors wrote, read, discussed, and approved the manuscript.

## Funding

## Competing interests

The authors declare no competing interests.

## Additional information

**Correspondence and requests** for materials should be addressed to Roderich D. Süssmuth.

