## [Peer Review File · Nature Communications]

The occurrence of ansamers in the synthesis of cyclic peptidesREVIEWER COMMENTS

Reviewer #1 (Remarks to the Author):

The article “The occurrence of ansamers in the synthesis of cyclic peptides” by Yao et al. describes the synthesis of amanitin analogues and their study using a host of experimental (spectroscopy, crystallography) and computational (molecular dynamics and quantum chemical) methods. The results revealed two isomers of amanitin, and consideration of their structure and of other, similar, cyclic molecules led to the suggestion of the term ‘ansamers’ to describe the configuration of stereoisomers of bridged cyclic systems and other conformationally restricted cyclic systems (seemingly instead of the existing term ‘atropisomers’).

This review focuses on the MD simulations presented by Yao et al. Molecules 4a and 4b and the precursor 3b were each simulated for 20 μs (20 replicate simulations of 1 μs each) at 300 K and 0.1 μs (one simulation of 0.1 μs) at 400 K and their conformational preferences analysed. These simulations are analysed in a variety of ways to provide structural insight into some of the experimental findings.

Overall, I appreciated the detailed descriptions of the computational methods – it’s seldom that I think I could reproduce some research based entirely on the Methods section of a published paper – and also the way in which the simulations were used alongside the experimental data to produce a more complete picture.

For instance, the experimental finding that precursor 3b exclusively yields 4a in the synthesis procedure was explained by showing that in the MD simulations, 3b predominantly occupies conformations in which the tryptathionine and the ring planes are positioned similarly to in 4a. It was not made clear, however, whether this structure would also ‘prime’ 3b for reaction to form 4a. Additionally, it is not clear how much these results may have been affected by the choice of initial structure for the simulations of 3b (see also my comments below on Figure S3(a)).

The MD simulations were also used to add weight to the idea that transition between 4a and 4b is sterically difficult. It should be noted, however, that the force field parameters used for the MD simulations could have been sufficient to prevent such transitions; only QM calculations could truly show whether this conversion is possible.

The interpretation of the hydrogen bonds in terms of how often the set of bonds formed in the crystal structure (set 1) versus an alternative and seemingly mutually exclusive set of hydrogen bonds (set 2) is hampered by not consistently using different colours to plot the occurrence of each set of hydrogen bonds (see detailed comments on Figure S11 below).

Comments on the main figures:

Figure 2(b): It is not described in the Methods (Article or Supporting Information) how the highest-probability structure was obtained.

Figure 2(c): It would be more helpful to state that the colouring of the lines representing each plan is as per parts (a) and (b) of this figure, rather than referring to Figure 1, or at least refer more specifically to Figure 1(d), (e) or (g).

Figure 3(e): Why are the hydrogen bond schematics drawn differently for the X-ray crystal and simulation-derived data (i.e., purely vertical/horizontal dashed lines for the experimental structure vs diagonal lines for the computational data; amide groups only shaded for the X-ray crystal data)? This is not a big problem with respect to interpretation, it just seems oddly inconsistent.

Comments on the supplementary data:

Figure S2: A colour scale bar is needed for the Ramachandran plots in part (a) of this figure.

Figure S3(a): The RMSF values for 4a and 4b could be computed with respect to their respective crystal structures. However, there is no experimental structure for 3b, so the RMSF values were calculated with respect to the initial structure of the simulation. Given that the crystal structures are likely low-energy, highly-populated structures, what evidence is there that the structure of 3b used to initiate the simulations is comparable (in terms of being highly favourable)? This is important for interpretation of the RMSF values. The simulation of 3b shows higher RMSF values than the simulations of 4a and 4b; this could be due to it being more flexible (expected as the 'grey' cycle is not formed), but it could also be due to use of an inappropriate reference structure.

Figure S3(a): Glycine is considered to have a single-atom side chain (H). Why are there two atoms indicated as being in the side chain of each of the glycine residues (Gly5 and Gly7)?

Figure S11: The colours in which some of the hydrogen bond time-series are plotted appear to be wrong. e.g., the hydrogen bond time-series for starting structure 1 (SS1) begins with [0,2] and ends with [0,1] but is coloured orange, whereas according to the key, it should be purple and then cyan? I can imagine that orange has been used here and elsewhere because it would be a little fiddly to plot the two sections of the time-series separately. But then what about SS7? Surely it should be cyan? Using separate colouring for instances of sets 1 and 2 would also make it easier to see how often each is formed, and thus how often the hydrogen bonds formed in the MD simulations agree with those formed in the crystal structure (required to support statements such as "For 4a, in the majority of trajectories, the hydrogen bond pattern matches the crystal structure (Fig. S11).")

Figure S13/NOE data: Did the authors consider back-calculating the NOE contacts from the simulations to compare with the experimental data shown in Figure S13?

Table S4: It is stated in the caption that "The time points are determined by visual inspection of Figures S11-12 and S16." Figure S12 shows no time data and Figure S16 does not exist. Additionally, why not use the actual time-series data plotted in Figure S11 rather than visually inspect the graph? Was it really possible to come up with such specific time values as 0.0013 μ s by visual inspection?

Table S5: How do the eight populations of hydrogen bonds identified in this table correspond to the two hydrogen bond sets in Table S4 and [ELSEWHERE]?

Supporting Information Section 4.1: The three peptides are first equilibrated under NVT and then NpT conditions at 400 K, and then each of the 20 replicates is further equilibrated under NVT and then NpT conditions at 300 K. However, throughout all of these equilibrations, the peptide atoms are restrained to their initial positions, and there is no equilibration without position restraints prior to beginning the production run. Can the authors comment on whether there was any bias observed in the production runs towards the initial conformations due to this lack of 'free' equilibration? Additionally, in selecting the 20 structures to initiate the 20 independent replicates, the RMSD is calculated "for each prerun". Does this mean for each of the NVT and NpT 400 K equilibrations? Or does the 'each' refer to the three peptide molecules that were simulated (4a, 4b and 3b)? Lastly, why was such a high simulation temperature of 400 K used for the initial equilibration simulations?

Supporting Information Section 4.1: The lists of atom indices in the order in which the RMSF values were plotted is not particularly helpful with respect to interpreting the RMSF data. How is the reader supposed to know which atom corresponds to which index? Much more helpful would be chemical structures with the atom numbers labelled, or PDB (or similar) formatted files that indicate both atom type and atom number, as well as improving the ability of other researchers to replicate the work described here.

Minor typographical issues:

Throughout, 'replica' is used where the plural ('replicas') is required.

Figure S11: In this figure caption, "...in each replica: only set 2 or noise,..." should I think read "...in each replica. Purple: only set 2 or noise,..."

Table S4: In this table caption, "Approximate time points for the transition from the starting structure to the crystal structure for in [μ s] for each replica of 4a." should I think read "Approximate time points for the transition from the starting structure to the crystal structure for each replica of 4a." (the time units are provided in the column headers). Additionally, the column heading 'Hydrogen bond sets' should not span the 'Transition' and first 't [μ s]' columns, only the first 't [μ s]' column.

Table S6: Should "amide functions" be "amide functional groups"?

Reviewer #2 (Remarks to the Author):

In this draft submitted by Süssmuth et al, the authors proposed the term "ansamer" for describing the atropisomers of bridged cyclic peptides through the authors' extensive efforts for synthesizing a-amanitin with the use of macrolactonization process for bicyclization at various positions. Among the attempted eight macrolactonization protocols, the six protocols afforded the natural-type atropisomer of a-amanitin. Other two macrolactonization gave non-natural-type atropisomer of a-amanitin ranging in a ratio of natural to nonnatural from 1:0.7 to 1:0.2. Until the authors' achievement, whether the atropisomers of a-amanitin really exists or not has remained to be controversial; however, this achievement proves the atropisomer to exist with the additional use of NMR, MD calculation and Marfey's test. The reaction for affording the natural isomer, the authors analyzed the possible structure involved in the cyclization by MD simulation of precursor 3b which gives only natural isomer. The reviewer feels that MD simulation of precursor 3c which affords non-natural isomer as main would increase the scientific significance of this manuscript. Additionally, the authors proposed the use of "ansamer" for describing the atropisomers of bridged cyclic peptides; therefore, additional application including synthetic works to other bicyclic peptides would be desired. However, the contents of this manuscript is high quality. From these point views, this manuscript will merit publication in chemistry-oriented high quality journal such as JACS, Angew, or Chem. Sci.

Additional Comment

Structure of Foigure 1(d) is correct? Is this non-natural form?

Reviewer #3 (Remarks to the Author):

This report by Süssmuth and co-authors describes the synthesis of amaninamide isomers and their careful structural analysis. The new term "ansamer" is introduced.

Compared to other substrates such as 2a, 3c provides 4b ansa-selectively with a ratio of 1:2.7. An X-ray of this second isomer was obtained (4a is described in ref.14).

Ansa compounds according to IUPAC are "Benzene derivatives having para positions (or meta) bridged by a chain (commonly 10 to 12 atoms long) (Latin ansa, handle). By extension, any arene bridged by a chain constrained to lie over one of the two faces of the arene."

A better terminology is clearly needed to describe the observed macrobicyclic isomers and the "handle" analogy would be indeed illustrative. However, the use of ansa according to this definition with a clear connection to (any) arene seems problematic (without redefining the IUPAC suggestion). For instance, the reference plane for "ansamers" would likely be expected to be an arene part and the orientation of the chain on one or the other face of the arene would provide a means to determine the configuration. The aim for a systematic nomenclature is certainly a good one, but another term is likely needed to better interconnect to other definitions.

The paper is overall interesting, otherwise well written and relevant. It is therefore recommended to reconsider this manuscript with the answers to the following points.

Suggestion points:

1. replace the term "ansamer" as discussed above
2. change "isomery" to "isomerism" (in all instances)
3. change "stereochemistry" to "configuration" (in all instances)
4. it would be useful to have the CCDC indicator for the structure 4a reported in ref. 14 directly in the manuscript
5. the conditions for the bicyclization could be directly given in Figure 2 (drawing & more details in the caption)
6. it would be interesting if different bicyclization reagents and conditions change selectivity
7. change "not prove racemization" to epimerization as the AAs (enantiomers) are only formed during analysis

Reviewer #1 (Remarks to the Author):

The article “The occurrence of ansamers in the synthesis of cyclic peptides” by Yao et al. describes the synthesis of amanitin analogues and their study using a host of experimental (spectroscopy, crystallography) and computational (molecular dynamics and quantum chemical) methods. The results revealed two isomers of amanitin, and consideration of their structure and of other, similar, cyclic molecules led to the suggestion of the term ‘ansamers’ to describe the configuration of stereoisomers of bridged cyclic systems and other conformationally restricted cyclic systems (seemingly instead of the existing term ‘atropisomers’).

This review focuses on the MD simulations presented by Yao et al. Molecules 4a and 4b and the precursor 3b were each simulated for 20 μ s (20 replicate simulations of 1 μ s each) at 300 K and 0.1 μ s (one simulation of 0.1 μ s) at 400 K and their conformational preferences analysed. These simulations are analysed in a variety of ways to provide structural insight into some of the experimental findings.

Overall, I appreciated the detailed descriptions of the computational methods – it’s seldom that I think I could reproduce some research based entirely on the Methods section of a published paper – and also the way in which the simulations were used alongside the experimental data to produce a more complete picture.

Response: We thank the reviewer for his/her encouraging words and for thoroughly reading the computational material. We reply to the questions and comments in detail below.

For instance, the experimental finding that precursor 3b exclusively yields 4a in the synthesis procedure was explained by showing that in the MD simulations, 3b predominantly occupies conformations in which the tryptathionine and the ring planes are positioned similarly to in 4a. It was not made clear, however, whether this structure would also ‘prime’ 3b for reaction to form 4a.

Response: Yes, we are certain that this structure primes 3b for reaction to form 4a. The relative orientation of the three rings in 4a is characterized by three angles between the ring planes (Fig. 2c, upper left panel). The distributions of these three angles fully coincide between 3b and 4a (purple and cyan curves in the other panels in 2c), whereas in 4b the three angles assume a very different conformation (orange curves in the other panels in 2c). Thus, the reaction 3b -> 4a is the only logical choice. Fig. 2c is based on a previous detailed study on the conformational preferences of the synthetic precursors and their propensity to react to either ansamer (Yao et al., Chemistry–A European Journal. 2019;25(34):8030-4).

Additionally, it is not clear how much these results may have been affected by the choice of initial structure for the simulations of 3b (see also my comments below on Figure S3(a)).

Response: Please see our answers to “Comments on the main figures” (further below).

The MD simulations were also used to add weight to the idea that transition between 4a and 4b is sterically difficult. It should be noted, however, that the force field parameters

used for the MD simulations could have been sufficient to prevent such transitions; only QM calculations could truly show whether this conversion is possible.

Response: *We are confident that the force-field captures the enormous sterical barrier between the two ansamers correctly.*

In classical force-fields the parameters that are involved in conformational changes (bonded and non-bonded parameters) are carefully parametrized against experimental data and data from QM calculations. An exception to this are the barriers for improper dihedral, which are used to enforce the configuration of a stereocenter. The target angle for such an improper dihedral is chosen based on QM calculations, but the barrier height is set to an arbitrarily high value. This is to ensure that the configuration of the stereocenter does not accidentally change during a long simulation.

In a recent (not yet published) study, we did the following experiment: we restrained one ansamer (bridge above the peptide ring) to the NOE-distance restraints obtained for the other ansamer (bridge below peptide ring). The simulation eventually fulfilled all distance restraints. However, this was not achieved by rotating the tryptathionine bridge below the peptide ring. Instead the stereocenter at the C α -atom of tryptophan was inverted from an L-amino acid to a D-amino acid. This shows that the steric barrier of the rotating the tryptathionine bridge is higher than the artificially high barrier of inverting the stereocenter. (Of course, once the restraints were released, the molecule relaxed from this highly strained structure to a different conformation.)

*Additionally, visual inspection of the amanitin structure in a molecular visualization program or using an actual molecular model kit shows that the indole moiety of the tryptophan side chain is too spacious to fit through the relatively small peptide ring. Experimentally we never observed an interconversion between **4a** and **4b**.*

The interpretation of the hydrogen bonds in terms of how often the set of bonds formed in the crystal structure (set 1) versus an alternative and seemingly mutually exclusive set of hydrogen bonds (set 2) is hampered by not consistently using different colours to plot the occurrence of each set of hydrogen bonds (see detailed comments on Figure S11 below).

Response *Thank you for pointing this out. We revised Figure S11, please see our detailed reply below.*

Comments on the main figures:

Figure 2(b): It is not described in the Methods (Article or Supporting Information) how the highest-probability structure was obtained.

Response: Thank you for pointing this out. We have added the following paragraph to our 'methods' section (see 4.2.9):

4.2.9 Extraction of highest-probability structures

Using the hydrogen bond time series described in 4.2.3.2, structures were extracted based on the linear correlation of certain hydrogen bonds.

For **3b**, the most-populated hydrogen bonds (on average referring to the entire data set) 'Asn1m-Trp4m' (60%), 'Trp4m-Asn1m' (37%), 'Gly7m-Trp4m' (51%) and 'Cys8m-Trp4m' (65%) are positively correlated (see Fig. S2). Hence, all frames were extracted, in which all four hydrogen bonds are present (28% of the entire data set). This sub-data set was checked based on the RMSD (see 4.2.5) of the backbone from Trp4 to Asn1 with a preceding least-squares fit on the existing ring, only. The RMSD of the sub-data set amounted to 0.05 nm on average with a standard deviation of 0.01 nm. Consequently, the sub-data set was considered unified and representative of the highest-probability structure.

Figure 2(c): It would be more helpful to state that the colouring of the lines representing each plan is as per parts (a) and (b) of this figure, rather than referring to Figure 1, or at least refer more specifically to Figure 1(d), (e) or (g).

Response: Thank you for pointing this out. We agree that it makes more sense to refer to Figure 2a and 2b to explain the color scheme. We revised the caption of Fig. 2c accordingly.

Figure 3(e): Why are the hydrogen bond schematics drawn differently for the X-ray crystal and simulation-derived data (i.e., purely vertical/horizontal dashed lines for the experimental structure vs diagonal lines for the computational data; amide groups only shaded for the X-ray crystal data)? This is not a big problem with respect to interpretation, it just seems oddly inconsistent.

Response: Thank you for the comment. We adapted the schematics in Figure 3e such, that all H-bonds are represented by the same line types but colored according to the origin of the data. The shading of amide groups represents the solvent accessibility as derived from VT NMR data and therefore, we only added it in the schematics showing experimental data. We have now added colored ellipses to the schematics of H-bond networks found in computational data that, analogously to the VT NMR data, show the accessibility of the respective amides as derived from calculated solvent accessible surface area (SASA) values (see revised Figure 3 below). We additionally updated Table S4 (see below) to include SASA data of the MD simulations and crystal structures of both ansamers, as well as the measured VT-NMR values, with the same color-coding as the amide groups.

Updated Figure 3

Updated Table S4.

	4a_MD (nm ²)	4a_cryst(nm ²)	4a_VT-NMR $\Delta\delta\text{HN}/\Delta\text{T}$ (ppbK ⁻¹)	4b_MD (nm ²)	4b_cryst (nm ²)	4b_VT-NMR $\Delta\delta\text{HN}/\Delta\text{T}$ (ppbK ⁻¹)
Total	10.039 ± 0.277	10.11		9.728 ± 0.25	9.257	
Asn¹	0.006 ± 0.008	0.005	-1.550	0.006 ± 0.007	0.0	-3.826
Ile³	0.005 ± 0.006	0.0	-1.793	0.007 ± 0.006	0.001	-6.421
Trp⁴	0.006 ± 0.008	0.0	-2.910	0.033 ± 0.017	0.027	-4.807
Gly⁵	0.015 ± 0.015	0.0	-1.432	0.072 ± 0.038	0.06	-4.962
Ile⁶	0.039 ± 0.028	0.063	-4.171	0.003 ± 0.008	0.0	0.287

Gly ⁷	0.064 ± 0.03	0.131	-3.403	0.016 ± 0.025	0.036	nd
Cys ⁸	0.023 ± 0.024	0.031	-2.358	0.001 ± 0.005	0.0	-2.686

Comments on the Supplementary Data:

Figure S2: A colour scale bar is needed for the Ramachandran plots in part (a) of this figure.

Response: *Thank you for spotting this, we added a color scale bar to Figure S2a.*

Figure S3(a): The RMSF values for 4a and 4b could be computed with respect to their respective crystal structures. However, there is no experimental structure for 3b, so the RMSF values were calculated with respect to the initial structure of the simulation. Given that the crystal structures are likely low-energy, highly-populated structures, what evidence is there that the structure of 3b used to initiate the simulations is comparable (in terms of being highly favourable)? This is important for interpretation of the RMSF values. The simulation of 3b shows higher RMSF values than the simulations of 4a and 4b; this could be due to it being more flexible (expected as the 'grey' cycle is not formed), but it could also be due to use of an inappropriate reference structure.

Response: *Thank you for pointing this out. We have now chosen one structure out of the subset containing the highest-probability structure of the MD ensemble of 3b and used it as reference structure for RMSF calculations. We adapted Figures S2a, S3a and S3b accordingly. In addition, the dihedral angles of the reference structure are indicated in the Ramachandran plot as blue crosses (Fig. S2a). The extraction of the highest-probability structure is described in Methods Section 4.2.9 'Extraction of highest probability structures'.*

Figure S3(a): Glycine is considered to have a single-atom side chain (H). Why are there two atoms indicated as being in the side chain of each of the glycine residues (Gly5 and Gly7)?

Response: *Thank you for your comment. In all our simulations, the 'main chain' is defined as 'N', 'H' (amide), 'CA', 'C' and 'O' for each residue - following the GROMACS convention. All remaining atoms of the respective residue are assigned to the 'side chain'. In this convention, H-atoms linked to the C_alpha are not considered as part of the main chain for any amino acid. Since the side chain of glycine is an H-atom, glycine has two H-atoms attached to the C_alpha atom and therefore two atoms are shown as 'side chain' in Figure S3a. We clarified this in the caption of Figure S3a.*

Figure S11: The colours in which some of the hydrogen bond time-series are plotted appear to be wrong. e.g., the hydrogen bond time-series for starting structure 1 (SS1) begins with [0,2] and ends with [0,1] but is coloured orange, whereas according to the key, it should be purple and then cyan? I can imagine that orange has been used here and elsewhere because it would be a little fiddly to plot the two sections of the time-series separately. But then what about SS7? Surely it should be cyan? Using separate colouring for instances of sets 1 and 2 would also make it easier to see how often each is formed, and thus how often the hydrogen bonds formed in the MD simulations agree with those formed in the crystal structure (required to support statements such as "For 4a, in the majority of trajectories, the hydrogen bond pattern matches the crystal structure (Fig. S11).")

Response: *Thank you for your comment. To improve the legibility of the figure, we added a panel to Figure S11 where we define the two states with their exclusive H-bond sets (S11a). Here, the H-bond set corresponding to state 1 is shown in olive and the H-bond set*

corresponding to state 2 is shown in blue. The same colors are used to present each state in the plots of the time series. We also added an inset showing a zoomed-in view with 2.5 ns of a trajectory to better illustrate that all states (in this example 0 and 1) are exclusive to each other (please see revised Figure S11 below).

Revised Figure S11

Figure S13/NOE data: Did the authors consider back-calculating the NOE contacts from the simulations to compare with the experimental data shown in Figure S13?

Response: *We thank the reviewer for their suggestion and expanded Figure S13 with two additional panels to illustrate the (1) agreement between the proton distances in the MD data and the NMR NOE data and (2) agreement between the proton distances in the MD data and the distances in the crystals for each isomer (please see revised Figure S13 below). Back-calculating NOE contacts is hampered by the high complexity of the simulation data (20 million structures) and we therefore considered NOE violations instead, a well-established method to compare NMR and MD data. For both isomers the agreement between the simulations and experimental data is very good with most distances deviating from each other by less than one Ångström:*

4a: *(NMR vs. MD) 94% and (crystal vs. MD): 95% of the considered distances were < 1Å*

4b: *(NMR vs. MD) 92%, (crystal vs. MD): 97% of the considered distances were < 1Å*

Revised Figure S13 Comparison of isomer NOE data and agreement with MD simulations. Distinct NOE patterns in contact maps of a) 4a and b) 4b. Distance violations of proton distances in the MD simulations compared to the NMR NOE data or crystal structures of c) 4a and d) 4b. The same set of proton-proton distances shown in a) and b) was used to calculate the violations.

Table S4: It is stated in the caption that “The time points are determined by visual inspection of Figures S11-12 and S16.” Figure S12 shows no time data and Figure S16 does not exist. Additionally, why not use the actual time-series data plotted in Figure S11 rather than visually inspect the graph? Was it really possible to come up with such specific time values as 0.0013 μ s by visual inspection?

Response: *Thank you for spotting this mistake. We did not visually determine the transition times between states 1 and 2 of isomer 4a as mistakenly stated in the caption of Table S4 but calculated them as described in Methods Section 4.2.10 ‘Determining transition times’. These transition times correspond to the trajectories shown in Figures S11a and S11b. The reference to figure S16 is an error due to a change in SI figure numbers, thank you for pointing it out.*

Table S5: How do the eight populations of hydrogen bonds identified in this table correspond to the two hydrogen bond sets in Table S4 and [ELSEWHERE]?

Response: *Table S4 and Figure S11 show data related to H-bonds observed in simulations of isomer 4a. Table S5 lists the various combinations of H-bonds that we observed in our simulations of isomer 4b. However, we found that the data provided in Table S5 was not needed and to avoid confusion we removed the table from the revised “Supplementary Information”.*

Supporting Information Section 4.1: The three peptides are first equilibrated under NVT and then NpT conditions at 400 K, and then each of the 20 replicates is further equilibrated under NVT and then NpT conditions at 300 K. However, throughout all of these equilibrations, the peptide atoms are restrained to their initial positions, and there is no equilibration without position restraints prior to beginning the production run. Can the authors comment on whether there was any bias observed in the production runs towards the initial conformations due to this lack of ‘free’ equilibration? Additionally, in selecting the 20 structures to initiate the 20 independent replicates, the RMSD is calculated “for each prerun”. Does this mean for each of the NVT and NpT 400 K equilibrations? Or does the ‘each’ refer to the three peptide molecules that were simulated (4a, 4b and 3b)? Lastly, why was such a high simulation temperature of 400 K used for the initial equilibration simulations?

Response: *Thank you for your questions. The general procedure for simulating each of the three peptides was as follows (short summary of section 4.1.2):*

pre-run at 400K: *energy minimisation of a single solvated system \rightarrow NVT, NPT equilibrations with position restraints at 400 K \rightarrow simulation for 100 ns at 400 K without any restraints*

extraction of 20 diverse replica and equilibration: *Calculation of the all-atom RMSD and dividing-up the distribution of the RMSD in 20 equally-sized bins \rightarrow extraction of one random structure out of each bin, in total: 20 structures - each considered to be the starting structure of an own replica \rightarrow NVT, NPT equilibrations at 300 K with position restraints on the peptide for all 20 replicas*

production run at 300K: *simulation without any restraints for all 20 replicas.*

Due to the tryptathionine bridge, the conformational flexibility of amanitin is very limited. It is not certain that the peptide can visit a conformation that markedly differs from the crystal structure. If such a conformation exists, then it is likely separated by a relatively high barrier from the crystal structure. The system set-up as described above serves two purposes:

- conformational search at 400 K to identify as many diverse conformations as possible.*
- equilibration of the identified conformations for a simulation at 300 K*

For 4a, we in fact find a conformation that differs from the crystal structure (state 2 in Fig S11). This conformation is stable on the timescale of several 100 ns before it converts into state 1. We do not observe any transitions from state 1 to state 2 in our simulations.

For 4b, we find only a single structure defined by three hydrogen bonds (see Figs 3c and S3b). Occasionally, these hydrogen bonds fluctuate across the structural boundaries that we used to identify the hydrogen bonds. But only in 2% of all frames none of the three hydrogen bonds is formed.

We first keep the position constraints on during the equilibration at 300 K in order to ensure that the structure identified during the conformational search does not immediately relax to the crystal structure. In principle, this should be followed by a simulation without position restraint of 500 ps to 1 ns to further equilibrate the system before the actual production run starts. This equilibration should then be excluded from the analysis. Here, we subsume this equilibration into our production run. For a simulation of 20 μ s, 1 ns equilibration time represents only 0.005% of the total data. Thus, the error we make by not excluding the equilibration from the analysis is negligible.

Supporting Information Section 4.1: The lists of atom indices in the order in which the RMSF values were plotted is not particularly helpful with respect to interpreting the RMSF data. How is the reader supposed to know which atom corresponds to which index? Much more helpful would be chemical structures with the atom numbers labelled, or PDB (or similar) formatted files that indicate both atom type and atom number, as well as improving the ability of other researchers to replicate the work described here.

Response: *Since the RMSF was calculated with reference to the respective crystal structure (4a, 4b) or the lowest energy structure of the MD simulations (3b), the atom indices refer to the numbering of atoms in the structure files. We have now included the structure files (.pdb) of the reference structures for all three molecules in the supplementary material. We also updated the caption of Figure S3 to clarify the order in which the atoms are presented.*

Minor typographical issues:

Throughout, 'replica' is used where the plural ('replicas') is required.

Response: *Thank you. We corrected this mistake.*

Figure S11: In this figure caption, "...in each replica: only set 2 or noise,..." should I think read "...in each replica. Purple: only set 2 or noise,..."

Response: *Thank you. We revised the caption of Figure S11.*

Table S4: In this table caption, “Approximate time points for the transition from the starting structure to the crystal structure for in [μ s] for each replica of 4a.” should I think read “Approximate time points for the transition from the starting structure to the crystal structure for each replica of 4a.” (the time units are provided in the column headers). Additionally, the column heading ‘Hydrogen bond sets’ should not span the ‘Transition’ and first ‘t [μ s]’ columns, only the first ‘t [μ s]’ column.

Response: *Thank you. We revised Table S4 and corrected the caption.*

Table S6: Should “amide functions” be “amide functional groups”?

Response: *Thank you, we corrected this.*

Reviewer #2 (Remarks to the Author):

In this draft submitted by Sussmuth et al, the authors proposed the term “ansamer” for describing the atropisomers of bridged cyclic peptides through the authors’ extensive efforts for synthesizing a-amanitin with the use of macrolactonization process for bicyclization at various positions. Among the attempted eight macrolactonization protocols, the six protocols afforded the natural-type atropisomer of a-amanitin. Other two macrolactonization gave non-natural-type atropisomer of a-amanitin ranging in a ratio of natural to nonnatural from 1:0.7 to 1:0.2. Until the authors’ achievement, whether the atropisomers of a-amanitin really exists or not has remained to be controversial; however, this achievement proves the atropisomer to exist with the additional use of NMR, MD calculation and Marfey’s test. The reaction for affording the natural isomer, the authors analyzed the possible structure involved in the cyclization by MD simulation of precursor **3b** which gives only natural isomer. The reviewer feels that MD simulation of precursor **c** which affords non-natural isomer as main would increase the scientific significance of this manuscript. Additionally, the authors proposed the use of “ansamer” for describing the atropisomers of bridged cyclic peptides; therefore, additional application including synthetic works to other bicyclic peptides would be desired. However, the contents of this manuscript is high quality. From these point views, this manuscript will merit publication in chemistry-oriented high-quality journal such as JACS, Angew, or Chem. Sci.

Response: *Thank you for the constructive comments. We set up MD simulations of **3c** and analyzed the data. Like for **3b**, we extracted the highest-probability structure by extracting all frames, in which the most-populated, positively correlated hydrogen bonds are present. For this subset, we analyzed the RMSD on all atoms and on the backbone atoms of the existing ring. The RMSD showed that although there is little flexibility in the backbone of the existing ring (RMSD < 0.04 nm), the loose N- and C-termini can vary substantially in their positions (RMSD all atoms = 0.44 nm). In contrast to **3b**, the highest-probability structure is therefore not that clear. Besides, in all structures of this subset, we observe a structural arrangement, in which the tryptathionine bridge is similarly positioned as in **4a**. Apart from the highest-probability structure, we also calculated the RMSD of the MD data set of **3c** towards the crystal structures to see whether structures are formed that show resemblance to one of the crystal structures. We observe a broad distribution over all replicas ranging*

from 0.3 to 0.8 nm (see figureA below), thus **3c** does not show a clear predisposition for any isomer.

Figure S18: Average distributions of the all-atom RMSD (after least-square fit on the backbone atoms of the existing ring) over all 20 replicas for each MD simulation data set (as labelled: top left : **4a**, bottom left : **4b**, top right : **3b** and bottom right : **3c**) towards different reference structures: 'purple' : **4a**, 'orange' : **4b**, 'green' : **3b**, 'blue' : **3c**. For **4a** and **4b**, the crystal structures were used as reference structures. For **3b** and **3c** the highest-probability structures were used.

Additional Comment

Structure of Figure 1(d) is correct? Is this non-natural form?

Response: The structural formula of Figure 1(d) is alpha-amanitin. The 2D structure is not able to distinguish the natural ansamer or non-natural ansamer.

Reviewer #3 (Remarks to the Author):

This report by Süssmuth and co-authors describes the synthesis of amaninamide isomers and their careful structural analysis. The new term "ansamer" is introduced. Compared to other substrates such as **2a**, **3c** provides **4b** ansa-selectively with a ratio of 1:2.7. An X-ray of this second isomer was obtained (**4a** is described in ref.14). Ansa compounds according to IUPAC are "Benzene derivatives having para positions (or meta) bridged by a chain (commonly 10 to 12 atoms long) (Latin ansa, handle). By extension, any arene bridged by a chain constrained to lie over one of the two faces of the arene." A better terminology is clearly needed to describe the observed macrobicyclic isomers and the "handle" analogy would be indeed illustrative. However, the use of ansa according to this definition with a clear connection to (any) arene seems problematic (without redefining the IUPAC suggestion). For instance, the reference plane for "ansamers" would likely be expected to be an arene part and the orientation of the chain on one or the other face of the arene would provide a means to determine the configuration. The aim for a systematic nomenclature is certainly a good one, but another term is likely needed to better interconnect to other definitions.

Response: *Thank you for your valuable comments. Indeed, before submitting the manuscript for review we were discussing these issues for quite some while in the team. Here are our thoughts:*

1. The term atropisomer: *The here presented structures are characterized by a handle-type bridge, which can be moved in and out of the cyclic peptide structure. While atropisomers can be interconverted by rotation of connecting moieties, which are typically sterically hindered, ansa-compounds (lat. ansa = handle), e.g. cyclophanes, are interconverted by rotation of the handle around the planar phenylene. The bridged cyclic peptide structures cannot be appropriately described as atropisomers, because the interconversion of these stereoisomers can not be attributed to the rotation around a hindered single bond.*
2. The term ansamer: *As a consequence of above considerations, we propose the term ansamer to describe (peptide) stereoisomers with a handle, which determines the configuration of these stereoisomers. We understand the objections of the referee, but feel on the other hand that the intended similarity between „ansa compound“ for aromates and „ansamer“ will raise attention and spur discussions in the scientific community about the underlying problem. It also shows that the previously chosen term on “ansa compounds” was probably chosen in a too restricted way and that the present examples require a discussion in a wider context. Alternative suggestions, e.g. a construction with the Greek word for handle, are unhandy/uneasy to spell and, to our estimate, will be neglected. Therefore, we would like to stay with this expression until further knowledge about the topic is gathered to make a final suggestion on nomenclature.*

The paper is overall interesting, otherwise well written and relevant. It is therefore recommended to reconsider this manuscript with the answers to the following points.

Suggestion points:

1. replace the term "ansamer" as discussed above

Response: *Thank you for the comments. See comprehensive answer given above.*

2. change "isomery" to "isomerism" (in all instances)

Response: *Thank you for the comments, we have replaced the term "isomery" with "isomerism" in entire manuscript.*

3. change "stereochemistry" to "configuration" (in all instances)

Response: *Thank you for the comments, we have replaced "stereochemistry" with "configuration" in the entire manuscript.*

4. it would be useful to have the CCDC indicator for the structure 4a reported in ref. 14 directly in the manuscript

Response: *Thank you for the comment, we completely agree with this and have inserted the CCDC indicator of **4a** in the manuscript.*

5. the conditions for the bicyclization could be directly given in Figure 2 (drawing & more details in the caption)

Response: *Thank you for pointing this out, we have added the bicyclization condition in Figure 2 and the short description in the caption.*

6. it would be interesting if different bicyclization reagents and conditions change selectivity

Response: *Thank you for the comment. For the bicyclization of **2b**, the coupling reagent HATU was initially tested but significant amounts of the guanidination product were detected. The ratio of natural isomer and non-natural isomer is approximately 1:0.7. To suppress the formation of the guanidination product, EDC/HOAt was selected as coupling reagent. This also resulted in a ratio of natural isomer and non-natural isomer of 1:0.7 (see figure below, which has been implemented as Figure S16 into the manuscript). So, we proposed that different bicyclization reagents and conditions do not significantly change the ansa-selectivity of the reaction.*

Figure S16 Formation of ansamers in different coupling reagents. HPLC chromatograms of the products of the coupling reaction in HATU (blue line) and EDC/HOAt (pink line) show similar isomer ratios and a peak for the guanidination product in HATU.

7. change "not prove racemization" to "epimerization" as the AAs (enantiomers) are only formed during analysis

Response: Thank you for the comment, we have changed the sentence according to the reviewer's suggestion.

REVIEWER COMMENTS

Reviewer #1 (Remarks to the Author):

The authors have addressed all of the points from my review. In my opinion, the computational aspects of the manuscript (upon which my review focused) are suitable for publication without further changes.

Reviewer #2 (Remarks to the Author):

Sussmuth et al, proposed the term "ansamer" for describing the atropisomers of bridged cyclic peptides through the authors' extensive efforts for synthesizing α -amanitin with the use of macrolactonization process for bicyclization at various positions and for carrying out detailed MD simulation works. As a response to the reviewers' comments, the authors responded well. The reviewer feels that this revised version should merit publication in Nature Communication.

Reviewer #3 (Remarks to the Author):

The authors addressed all points raised by the reviewers

Reviewer #4 (Remarks to the Author):

The underlying result of the work described in this paper is the observation that altering the specific location of the newly-formed bond in the formation of a cross-linked cyclic peptide drug from 2 components can result in the production of two alternative conformers which are not interchangeable due to steric considerations and are thus atropisomers. This result is interesting and valuable as it demonstrates the potential limitation of available synthetic routes for such compounds so that the correct conformer is synthesised. The structural characterisation of the new conformer and differences from existing "natural" structure is well characterised and supported by evidence from a number of orthogonal methodologies (crystallography, NMR, chromatography). Molecular dynamics simulations support the lack of convertibility between forms, and support the crystal structures as being close to the population of lowest energy structures of the molecules. I have no substantial criticism of this aspect.

The authors then seek to place this kind of pseudo-isomerism in the context of related work, and to coin the term "ansamers" for these pairs of atropisomers. While I understand the basis for this choice, I am not convinced that this is a good choice, and that it may cause unnecessary confusion. Ansa compounds form pairs of true stereoisomers (enantiomers). The reference to "ansa" in ansamers may confuse this concept as the pairs of ansamers are only mirror images in the sense of their gross topology. It is in keeping with the major works of Prelog et al and Nicolaou et al (which the authors cite), that the P/M helical descriptors are appropriate for these molecules, as the authors use. But, the particular example of ansa stereochemistry is perhaps not the most helpful or intuitive. Have the authors considered a step back to considering P/M helical geometry, and a formulation as they show in Fig 2c where the axis that is being tested for P/M is the bridgehead-bridgehead axis. Both $\phi(\text{AC})$ and $\phi(\text{CB})$ in fig 2c yield a "P" for compound 4a for example. The extension to the other types of atropisomerism (lasso, norborino etc) may also be confused by the "ansa" terminology, while the P/M classification is also in principle uncontroversially applicable to them.

REVIEWER COMMENTS

Reviewer #1 (Remarks to the Author):

The authors have addressed all of the points from my review. In my opinion, the computational aspects of the manuscript (upon which my review focused) are suitable for publication without further changes.

Reply: We thank the reviewer for the positive assessment and the appreciation of our manuscript.

Reviewer #2 (Remarks to the Author):

Sussmuth et al, proposed the term “ansamer” for describing the atropisomers of bridged cyclic peptides through the authors’ extensive efforts for synthesizing amanitin with the use of macrolactonization process for bicyclization at various positions and for carrying out detailed MD simulation works. As a response to the reviewers’ comments, the authors responded well. The reviewer feels that this revised version should merit publication in Nature Communication.

Reply: We thank the reviewer for the positive assessment and the appreciation of our comments.

Reviewer #3 (Remarks to the Author):

The authors addressed all points raised by the reviewers

Reply: We appreciate reviewer #3 for the careful reading of our manuscript and we are glad that we had answered all questions that reviewer #3 concerned.

Reviewer #4 (Remarks to the Author):

The underlying result of the work described in this paper is the observation that altering the specific location of the newly-formed bond in the formation of a cross-linked cyclic peptide drug from 2 components can result in the production of two alternative conformers which are not interchangeable due to steric considerations and are thus atropisomers. This result is interesting and valuable as it demonstrates

the potential limitation of available synthetic routes for such compounds so that the correct conformer is synthesised. The structural characterisation of the new conformer and differences from existing "natural" structure is well characterised and supported by evidence from a number of orthogonal methodologies (crystallography, NMR, chromatography). Molecular dynamics simulations support the lack of convertibility between forms, and support the crystal structures as being close to the population of lowest energy structures of the molecules. I have no substantial criticism of this aspect.

Reply: Thank you for your valuable comments. Indeed, before submitting the manuscript for review we were discussing these issues for quite some while in the team.

The authors then seek to place this kind of pseudo-isomerism in the context of related work, and to coin the term "ansamers" for these pairs of atropisomers. While I understand the basis for this choice, I am not convinced that this is a good choice, and that it may cause unnecessary confusion. Ansa compounds form pairs of true stereoisomers (enantiomers). The reference to "ansa" in ansamers may confuse this concept as the pairs of ansamers are only mirror images in the sense of their gross topology. It is in keeping with the major works of Prelog et al and Nicolaou et al (which the authors cite), that the P/M helical descriptors are appropriate for these molecules, as the authors use. But, the particular example of ansa stereochemistry is perhaps not the most helpful or intuitive. Have the authors considered a step back to considering P/M helical geometry, and a formulation as they show in Fig 2c where the axis that is being tested for P/M is the bridgehead-bridgehead axis. Both $\phi(\text{AC})$ and $\phi(\text{CB})$ in fig 2c yield a "P" for compound 4a for example. The extension to the other types of atropisomerism (lasso, norborno etc) may also be confused by the "ansa" terminology, while the P/M classification is also in principle uncontroversially applicable to them.

Reply: We thank the reviewer for her/his insightful comments on the choice of the term "ansamers". We agree that the P/M terminology applied to helical geometry is basically also applicable for the isomer pairs we discuss in our manuscript. The choice was done since P/M is handy, robust and easy to grasp and in order not to confuse with the highly complex requirements for R/S nomenclature (CIP). Initially, we considered various approaches to assign a descriptor to these types of stereoisomers. It is important to note, that for a chirality axis 'M' = 'R' and 'P' = 'S',

for chirality planes the opposite relationship is 'M' = 'S' and 'P' = 'R'. Instead of considering the bridgehead-bridgehead axis as suggested by the reviewer, we developed a system which is also consistent with chemical nomenclature, namely, that the main ring is the basis, in this case the planar system. Another reason is, that if we would apply the descriptor as suggested, there would be no difference in the nomenclature between the two diastereomers **4a** and **4b** (bridge up or down).

The isomers **4a/4b** are not enantiomers, but diastereomers, not only because of the pronounced differences in bond geometries, but also due to the chirality of amino acids. In our opinion, however, it is even more controversial if "atropisomers" is an accurate term for the observed isomerism? Generally, atropisomers arise due to restricted rotation around a single bond (axis) which is not the case here. It is clear that the "bridge up" and "bridge down" amanitins are configurational isomers with the same molecular formula and same bond connectivities and we find that they cannot readily interconvert.

At this time there is no accurate and simple term to describe such a pair of isomers and, after long considerations, we would like to suggest "ansamers". Inspired by the terminology used for ansa compounds, the term "ansa" (handle) illustrates the source/reason of the isomerism and at the same time reflects the planar aspect shared with benzene ansa compounds. In this sense the nomenclature is consistent. This suggestion is made in full consciousness that the scientific community would later make revisions and come up with a different suggestion. For now we want to raise attention and spur further discussions. Therefore, we would be happy if the reviewer could nevertheless consent in our suggestion and support publication.